# 4-Phenyl-butyric Acid Inhibits Japanese Encephalitis Virus Replication via Inhibiting Endoplasmic Reticulum Stress Response

**DOI:** 10.3390/v15020534

**Published:** 2023-02-14

**Authors:** Shuangshuang Wang, Keli Yang, Chang Li, Wei Liu, Ting Gao, Fangyan Yuan, Rui Guo, Zewen Liu, Yiqing Tan, Xianwang Hu, Yongxiang Tian, Danna Zhou

**Affiliations:** Key Laboratory of Prevention and Control Agents for Animal Bacteriosis (Ministry of Agriculture and Rural Affairs), Hubei Provincial Key Laboratory of Animal Pathogenic Microbiology, Institute of Animal Husbandry and Veterinary, Hubei Academy of Agricultural Sciences, Wuhan 430064, China

**Keywords:** Japanese encephalitis virus, 4-PBA, UPR signaling pathway, non-structural protein 5

## Abstract

Japanese encephalitis virus (JEV) infection causes host endoplasmic reticulum stress (ERS) reaction, and then induces cell apoptosis through the UPR pathway, invading the central nervous system and causing an inflammation storm. The endoplasmic reticulum stress inhibitor, 4-phenyl-butyric acid (4-PBA), has an inhibitory effect on the replication of flavivirus. Here, we studied the effect of 4-PBA on JEV infection both in vitro and vivo. The results showed that 4-PBA treatment could significantly decrease the titer of JEV, inhibit the expression of the JEV NS3 protein (in vitro, *p* < 0.01) and reduce the positive rate of the JEV E protein (in vivo, *p* < 0.001). Compared to the control group, 4-PBA treatment can restore the weight of JEV-infected mice, decrease the level of IL-1β in serum and alleviate the abnormalities in brain tissue structure. Endoplasmic reticulum stress test found that the expression level of GRP78 was much lower and activation levels of PERK and IRE1 pathways were reduced in the 4-PBA treatment group. Furthermore, 4-PBA inhibited the UPR pathway activated by NS3, NS4b and NS5 RdRp. The above results indicated that 4-PBA could block JEV replication and inhibit ER stress caused by JEV. Interestingly, 4-PBA could reduce the expression of NS5 by inhibiting transcription (*p* < 0.001), but had no effect on the expression of NS3 and NS4b. This result may indicate that 4-PBA has antiviral activity independent of the UPR pathway. In summary, the effect of 4-PBA on JEV infection is related to the inhibition of ER stress, and it may be a promising drug for the treatment of Japanese encephalitis.

## 1. Introduction

Japanese encephalitis (JE) is an acute and severe zoonotic infectious disease caused by the Japanese encephalitis virus (JEV), which induces encephalitis in human beings and reproductive disorders in pigs, and causes significant harm to public health and the pork industry. JEV has a positive single-stranded RNA genome, which is cleaved by viral proteases, producing viral structure proteins of C, PrM, E and non-structural proteins of NS1, NS2a, NS2b, NS3, NS4a, NS4b and NS5. Non-structural proteins of JEV are responsible for virus replication [1] and are components of the viral replication complex (VRC) formed on the Endoplasmic Reticulum (ER) [2]. The ER activates the ER stress response due to the accumulation of newly synthesized unfolded proteins. At this time, cells will produce a compensatory regulation mechanism to adjust the protein misfolding and homeostasis imbalance in the endoplasmic reticulum cavity, which is called unfolded protein response (UPR). After virus infection, the endoplasmic reticulum shows obvious proliferation and hypertrophy. A variety of pathogens interact with the endoplasmic reticulum. Among them, 35 animal viruses and one plant virus can induce endoplasmic reticulum chaperones and activate UPR, and most of them are RNA viruses [3]. JEV can activate three UPR signal paths. To eliminate pathogens, GRP78 interacts with the E protein of JE to activate the downstream XBP1 gene [4]. Endoplasmic reticulum stress plays an important role in eliminating foreign pathogens and inhibiting virus proliferation.

The aromatic fatty acid, 4-phenyl-butyric acid (4-PBA), inhibits histone deacetylase and regulates the chaperone effect [5], and can improve the folding ability of ER [6,7,8] by regulating the binding of the chaperone hydrophobic region with an unfolded protein. It is also used to treat FDA-approved urea cycle disorder [9]. In endoplasmic reticulum-related diseases, 4-PBA regulates the hydrophobic region of GRP78, inhibits downstream inflammation [10] and UPR activation [11] and reduces adipogenesis [12]. When the endoplasmic reticulum environment is disordered, it inhibits the pain response [13], reduces Ca^2+^ accumulation [14] and increases mitochondrial membrane potential (MMP) [15]. In other diseases, 4-PBA inhibits muscle growth and glomerular injury mediated by leptin [16], reduces cell apoptosis [17] and reduces spinal cord injury induced by endoplasmic reticulum stress [18]. In viral therapy, 4-PBA interferes with the assembly of the HCV replication complex [19] and can also reduce endoplasmic reticulum stress and autophagy reaction to affect the titer of IAV [20].

Our study evaluated the effects of 4-PBA on JEV replication and ERS, providing insights into the mechanism of 4-PBA treatment of JEV.

## 2. Materials and Methods

### 2.1. Virus, Cell and Animal

The JEV strain (HW1 Genbank is NC-001437.1) and mouse glial cells (BV2) were preserved by our laboratory. The experimental animals were SPF BALB/c female mice aged 5 weeks (Wuhan, China). BV2 cells were maintained in MEM supplemented with 10% fetal bovine serum (HyClone, Logan, UT, USA), 1% penicillin/streptomycin (HyClone, Logan, UT, USA) and 1% glutamine (HyClone, Logan, UT, USA). Cells were incubated at 37 ℃ with 5% CO_2_.

### 2.2. Infection and Treatment

BV2 cells grown to approximately 80–100% confluence were infected with JEV at a multiplicity of infection (MOI) of 1.5. After 2 h of adsorption, infected cells were maintained in MEM medium supplemented with 2% fetal bovine serum incubation, and 2.5 ug/mL of tunicamycin (Sigma-Aldrich, St. Louis, MO, USA). The 4-Phenylbutyric acid (MCE Inc., Bloomfield, NJ, USA) was added to the medium at a concentration of 1 mM/mL at 2 h post infection (hpi). The RNA, protein and supernatant were harvested to evaluate the TCID_50_ of JEV and GRP78 levels.

### 2.3. Plasmid Construction and Transfection

The eukaryotic expression recombinant plasmids of JEV NS3, NS4B and NS5 proteins were constructed based on pEGFP-N1 (primers shown in Table 1). The recombinant plasmid was transferred into competent Escherichia coli DH5a cells and was validated by sequencing. The plasmid was transfected into BV2 cells by using Lipofectamine 2000. The expression of NS3, NS4B and NS5 proteins were examined by western blot.

### 2.4. RNA Extraction and Quantitative Real-Time PCR

Total RNA was cleaved by Trizol reagent, and extracted by FastPure® Cell/Tissue Total RNA Isolation Kit V2 (Vazyme Biotech Co., Ltd, Nanjing, China). RNA concentration was detected by spectrophotometer. RNA was reverse-transcribed into cDNA using HiScript III All-in-one RT SuperMix (Vazyme Biotech Co., Ltd, Nanjing, China). ChamQ SYBR® qPCR Master Mix (Vazyme Biotech Co., Ltd, Nanjing, China) was used to perform qPCR.The procedure for qPCR reactions was 95 ℃ for 5 min (1 cycle), 95 ℃ for 30 s, annealing temperature for 30 s and 72 ℃ for 30 s (45 cycles). The results were normalized with β-actin by the ΔΔCT method and are depicted as fold change over mock-infected control. Primer sequences are provided in Table 2.

### 2.5. Immunoblots Analysis

The cells (5 × 10^5^ or 1 × 10^6^) were lysed with a RIPA Lysis Buffer (Beyotime, Shanghai, China) containing protease inhibitors. BCA (Beyotime, Shanghai, China) was used to determine the protein concentration. The protein was separated on 10% SDS-PAGE gel, transferred to PVDF membrane and sealed at room temperature with 5% skimmed milk powder in Tris-Buffered saline (TBS) for 1 h. Then the protein was incubated at 4 ℃ overnight with a primary antibody. The secondary antibody was incubated at room temperature for 2 h, and then the enhanced chemiluminescence (ECL) reagent was used for image capture.

### 2.6. In Vivo Experiment in Mice

JEV (10^6.5^ TCID_50_) was intracranially injected into the 5-week-old BALB/c mice. The 4-PBA diluent is recommended by the instructions. 4-PBA (80 mg/kg) is injected once before the challenge. After the challenge, 4-PBA is injected once a day for a total of 9 times. After the blood is collected from the eye vein of mice, the brain tissue is taken for pathological section, immunohistochemistry, Western blot and qPCR. Pathological section and immunohistochemistry experiment and result analysis are contracted by Wuhan Pinofei Biotechnology Co., Ltd., Wuhan, China.

### 2.7. Laser Confocal Microscope

The recombinant plasmid NS3, NS4b and NS5 were transfected into BV2 cells. Four hours after transfection, the control group without 4-PBA of 1 mM/mL and the treatment group were set, respectively, and the immunofluorescence experiment was conducted 12 h after the transfection. The detection method was introduced from [21], and the cell samples were observed under a laser scanning confocal microscope.

### 2.8. Statistical Analysis

The real-time PCR test data were used as the mean value ± Standard deviation (means), and the data were expressed by the percentage of the treated group relative to the untreated group. The protein results were analyzed by Image J. Immunohistochemistry was analyzed by Image-pro pus 6.0 (Media Cybernetics, Inc., Rockville, MD, USA), and three 200-fold visual fields were randomly selected from each section in each group to determine the unified positive standard of all photos. The cumulative positive optical density value (IOD) of each photo was obtained by analyzing each photo, and the average optical density was calculated. A one-way ANOVA test in GraphPad Prism (version 5.0) software was used to analyze the significant differences of the data. *p* < 0.05 (*), indicating significant difference; *p* < 0.01 (**) and *p* < 0.001 (***).

## 3. Results

### 3.1. 4-PBA Inhibits the Replication of JEV In Vitro

To investigate the effects of 4-PBA on JEV RNA replication, we detected the titer of JEV in BV-2 cells. The results showed that 4-PBA could significantly reduce the titer of JEV in BV2 cells at 6 hpi and 12 hpi (Figure 1A). The western blot results also showed that 4-PBA could significantly inhibit the expression of the JEV NS3 protein (GeneTex, Irvine, CA, USA) (Figure 1B).

### 3.2. 4-PBA Inhibits the Replication of JEV In Vivo

Mice were intracranially injected with JEV at a titer of 10^6.5^ TCID_50_ and then treated with the 4-PBA by intraperitoneal injection at a dose of 80 mg/kg. The sagittal plane of the mouse brains was collected for pathological section, immunohistochemistry and western blot. The results showed that 4-PBA treatment could significantly inhibit JEV titers, and the inhibition effect was positively correlated with the number of treatments (Figure 2A,D). In addition, 4-PBA treatment can significantly restore the body weight and brain structure damage of JEV-infected mice (Figure 2A,B). Pathological section results show that after JEV infection, the brain tissue structure is moderately abnormal, some neurons disappear, a large number of neurons in CA1 area degenerate, neurons can see fiber tangles and the number of glial cells becomes more. After 4-PBA treatment, brain tissue structure was slightly abnormal, individual nerve cells in the CA1 area degenerated and glial cells increased slightly. The serum IL-1β (Enzyme-linked Biotechnology Co., Ltd., Shanghai, China) level of JEV-infected mice was inhibited by 4-PBA treatment (Figure 2C).

### 3.3. 4-PBA Inhibits JEV-Induced Activation of the IRE1 and PERK Pathways

The 4-PBA is an inhibitor of ERS. BV2 cells at 2 h post infection (hpi) with JEV at MOI of 1.5 and 2.5 ug/mL of Tunicamycin; the positive control group was set up. 1 mM/mL of 4-Phenylbutyric acid (4-PBA) was added 2 h after JEV infection. Compared to that in the untreated cells, the protein levels of GRP78, sXBP1 and pEIF2α in JEV-infected cells treated with 4-PBA reduced (Figure 3A,B). Compared with that in the blank control group, the effect of TM was significant. Those results indicated that 4-PBA significantly inhibits JEV-induced activation of the IRE1 and PERK pathways. At the same time, we have confirmed that 4-PBA can inhibit the expression of the GRP78 protein and mRNA in brain tissue (Figure 3C,D).

### 3.4. 4-PBA Inhibits the Activation of ERS Induced by JEV Non-structural Proteins

In this experiment, we selected the most obvious JEV non-structural protein that can induce the endoplasmic reticulum stress response of BV2 cells for detection (provided by additional materials). The recombinant plasmid NS3, NS4b and NS5 were transfected into BV2 cells. Four hours after transfection, the control group without 4-PBA of 1 mM/mL and the treatment group were set, respectively. Compared to that in the untreated cells, the levels of GRP78, sXBP1 and pEIF2α in NS4b and NS5 transfected cells treated with 4-PBA reduced, but that in NS3 transfection cells showed a non-significant difference of GRP78 (Figure 4A,B). 4-PBA inhibits the non-structural protein of JEV to activate ERS through the PERK and IRE1 pathways.

### 3.5. 4-PBA Inhibits NS5 Expression In Vitro

Since 4-PBA can protect cells by reducing ER stress, we examined whether 4-PBA interfered with NS3, NS4b and NS5 replication by reducing ER stress. The recombinant plasmids NS3, NS4b and NS5 transfected into BV2 cells. Four hours after transfection, the control group and treatment group with and without 1 mM/mL 4-PBA were set, respectively. The results revealed that the NS5 protein level was down-regulated by 4-PBA treatment (Figure 5A,B). Further analysis revealed that NS5 gene expression in 4-PBA-treated cells was significantly decreased compared with that in untreated cells; the number of genes expressed varies from 12% to 5% (Figure 5C).

## 4. Discussion

Viruses had the ability to induce host cells stress responses, hijack autophagy and UPR mechanisms and promote host infection and virus replication. Continuous endoplasmic reticulum stress worsens normal cell activity and induces autophagy and apoptosis [22]. JEV uses ER as its replication site by destroying ER homeostasis and activating ERS, leading to a variety of common diseases, such as neurodegenerative diseases, kidney diseases and so on. Dengue virus (DENV), hepatitis C virus (HCV) and Japanese encephalitis virus (JEV) all belong to the Flaviviridae family. At present, many small molecule inhibitors can inhibit virus replication [23]. The inhibition of protein translocation from ER to the cytosol or inhibiting the ER chaperone GRP94 by small molecule compounds led to a significant decrease in DENV and ZIKV replication [24,25]. Moreover, 4-PBA, which disrupts virus-induced hypo-acetylation of histone H3K9/K14, inhibits African swine fever virus (ASFV) [26] replication and inhibits ZIKV replication via the UPR pathway [27]. The inhibition of HIV-1 Tat-induced UPR/ER stress by 4-PBA significantly alleviated astrocyte-mediated Tat neurotoxicity [28]. Similar to the above results, our results show that 4-PBA can significantly inhibit JEV replication on BV2 cells. In vivo, we also performed experiments on mice to find that after JEV brain infection in mice and 4-PBA was used, immunohistochemistry and western blot results showed that JEV replication was affected. In addition, 4-PBA could treat brain tissue damage caused by JEV infection, restore the body weight of mice, inhibit the expression of inflammatory factors in serum.

Glucose-regulated protein 78 (GRP78) is an ER chaperone protein that can restore protein misfolding [29]. UPR is monitored by three typical branches of ERS sensors, which are ER-mediated membrane-associated proteins: eukaryotic initiation factor 2 α Kinase (PERK), inositol requiring enzyme 1 (IRE1) and activating transcription factor 6 (ATF6) [30]. In the well-functioning and “pressure-free” ER, these three transmembrane proteins bind to chaperone BiP/GRP78 in the endoplasmic domain (the amino-terminal of IRE1 and PERK and the carboxyl-terminal of ATF6). The ERS can interfere with their viral replication through the UPR pathway [31,32,33]. Relevant research found that GRP78 is an important host protein for JEV effect and replication, and can help JEV to enter the cell [34]. GRP78 and RdRp enhance viral replication in mitochondria [35]. The Hrd1 complex composed of endoplasmic reticulum luminal lectin, chaperone, endoplasmic reticulum membrane protein and cytoplasmic protein is the key factor of flavivirus replication [36,37]. GRP78 is a key protein for JEV to enter host cells and replicate [34]. Our result 3 shows that JEV can regulate the expression of GRP78 in BV2 cells, and activate ER stress response through the PERK and IRE1 signaling pathways. Further, we verified that 4-PBA can inhibit JEV-activated endoplasmic reticulum stress and virus replication for the first time, and in vivo experiments in mice indicated that 4-PBA could inhibit the expression of the GRP78 protein and mRNA in tissues.

JEV can activate the IRE1 pathway through PrM, E, NS1, NS2a, NS2b and NS4b proteins, and only NS4B can induce PERK dimerization and apoptosis for JEV to active PERK pathways [38]. After examining the effects of all non-structural proteins on ERS, we found that NS3, NS4A, NS4B and NS5 had the most significant regulation on GRP78. For NS3 and NS5 are the key proteins constituting the virus replication complex, and NS4B is located in the endoplasmic reticulum, we detected the effect of 4-PBA on those three proteins separately. The results showed that 4-PBA could inhibit their activated PERK and IRE1 pathways.

As an endoplasmic reticulum stress inhibitor, 4-PBA can significantly inhibit the transformation of LC3I/LC3II in PEDV-ORF3 of cells [39]. α-glucosidase inhibitors can inhibit four serotypes of dengue virus infection by disrupting the folding of the structural proteins prM and E [40,41]. NO can cause oxidative stress, but inhibit dengue virus replication by inhibiting the activity of RdRp [42]. Antioxidants inhibit viral replication by reducing the formation of negative-strand RNA and the accumulation of uncovered positive-strand RNA [43]. N-(4-hydroxyphenyl) retinamide (4-HPR) inhibits viral replication by inhibiting the transcription of the NS5 protein of DENV, but not through the PERK signal pathway [44]. A novel small-molecule compound (AO13) can inhibit replication of HCV, and might target the NS5B RNA polymerase [45]. Compound 147 can correct protein misfolding of ER, and the antiviral activity independent of ATF6 induction [46]. It is suggested that the antiviral effect of small molecules is mainly based on its effect on viral proteins. Our results revealed that the level of JEV NS5 protein and expression in cells was significantly decreased by 4-PBA. So, we believe that 4-PBA can inhibit JEV-activated endoplasmic reticulum stress response and has antiviral activity independent of the UPR pathway.

Japanese encephalitis (JE), as a zoonosis with high mortality and widespread impacts, seriously endangers the economic benefits of pig breeding. In some Asian countries, the JEV vaccine is mandatory [47], but Australia has not yet obtained a licensed JEV vaccine for pigs [48]. Therefore, the research and development of therapeutic drugs for JE virus is meaningful. Studying the mechanism of 4-PBA treatment of JEV infection in host cells at the molecular level can provide a new therapeutic target for the future research and development of drugs for the treatment of Japanese encephalitis, and also provide a theoretical basis for the research and development of new vaccines.

## Figures and Tables

**Figure 1 viruses-15-00534-f001:**
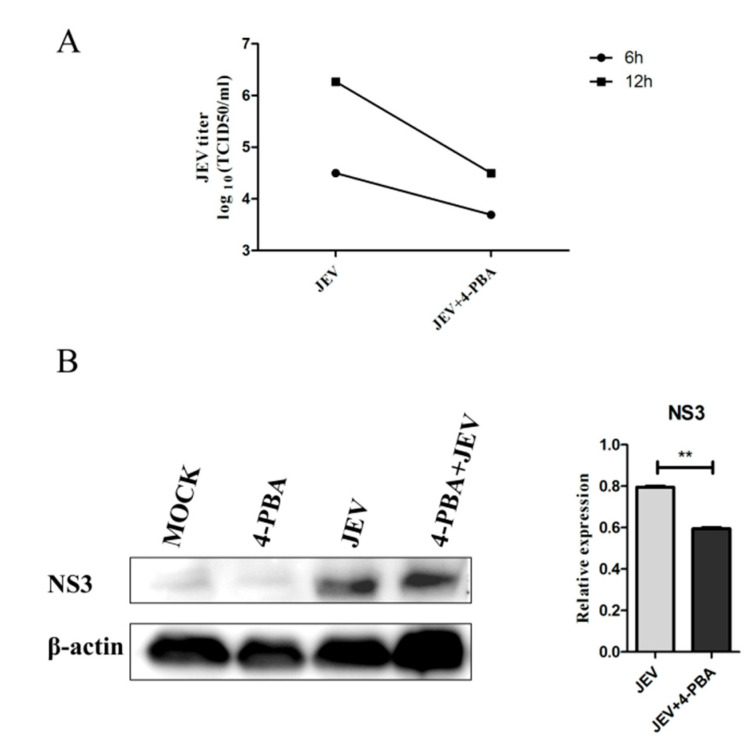
The titer of JEV virus decreased after treatment of 4-PBA. JEV (MOI 1.5) and 4-PBA of 1 mM/mL were used to treat BV2 cells. (**A**) JEV titers at 6 hpi and 12 hpi; the cell supernatant was measured by TCID50 assays. (**B**) The western blot at 12 h was used to test the NS3 protein expression of JEV. β-actin was used as sample loading control. ** *p* < 0.01.

**Figure 2 viruses-15-00534-f002:**
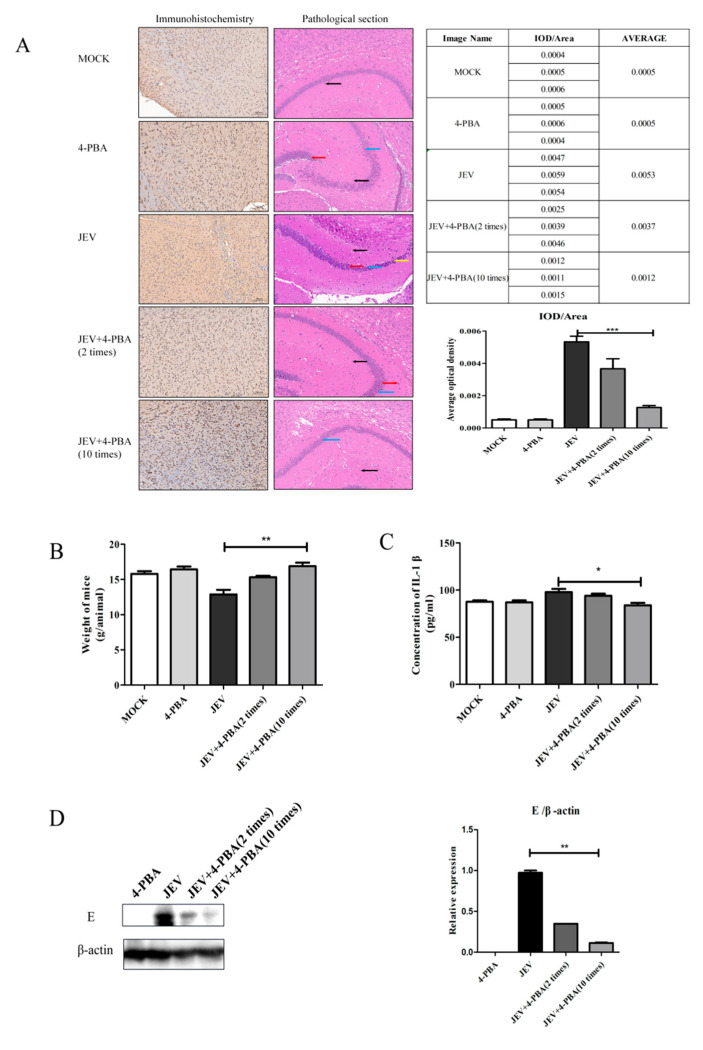
Therapeutic effect of 4-PBA in JEV-infected mice. (**A**) Detection of virus particles in brain tissues by immunohistochemistry and pathological analysis on sagittal plane of brain tissue. (**B**) Weight changes in mice infected with JEV and treated with 4-PBA. (**C**) Detection of IL-1β in serum. (**D**) Detection of virus particles in brain tissues by western blot. Yellow arrow: neurons disappear; blue arrow: a large number of nerve cells degenerate; red arrow: nerve fiber tangle; black arrow: glial cell. * *p* < 0.05; ** *p* < 0.01; *** *p* < 0.001.

**Figure 3 viruses-15-00534-f003:**
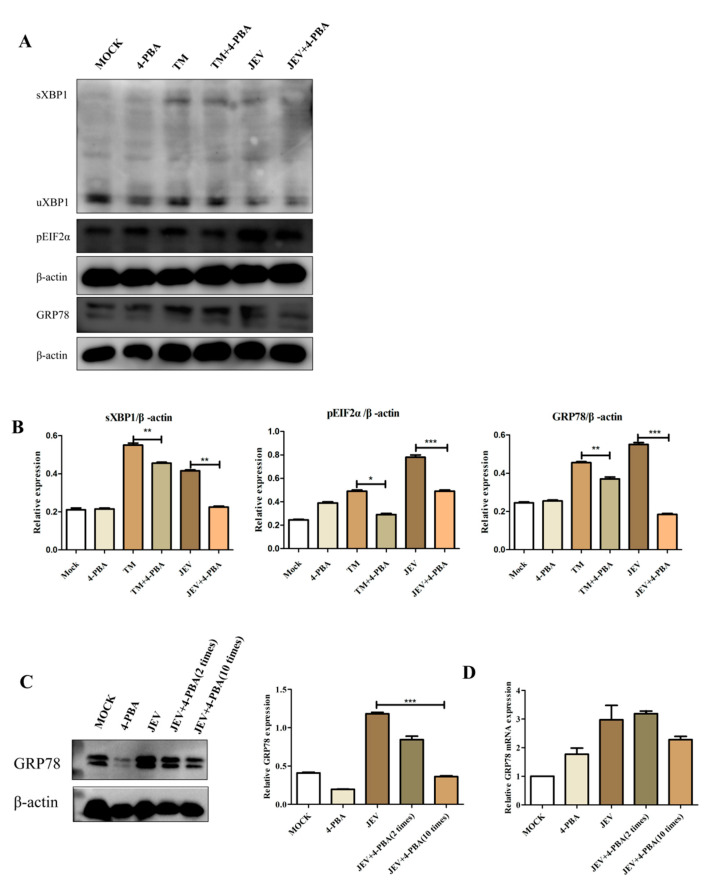
4-PBA inhibits JEV-induced activation of the IRE1 and PERK pathways. (**A**) The protein levels of GRP78, pEIF2α, sXBP1 (Abcam, Cambridge, UK) were determined using western blot. (**B**) Densitometry scans of the GRP78, pEIF2α and sXBP1 band in (**A**). (**C**) We have confirmed that 4-PBA can inhibit the expression of GRP78 mRNA and protein (**D**) in brain tissue. * *p* < 0.05; ** *p* < 0.01; *** *p* < 0.001.

**Figure 4 viruses-15-00534-f004:**
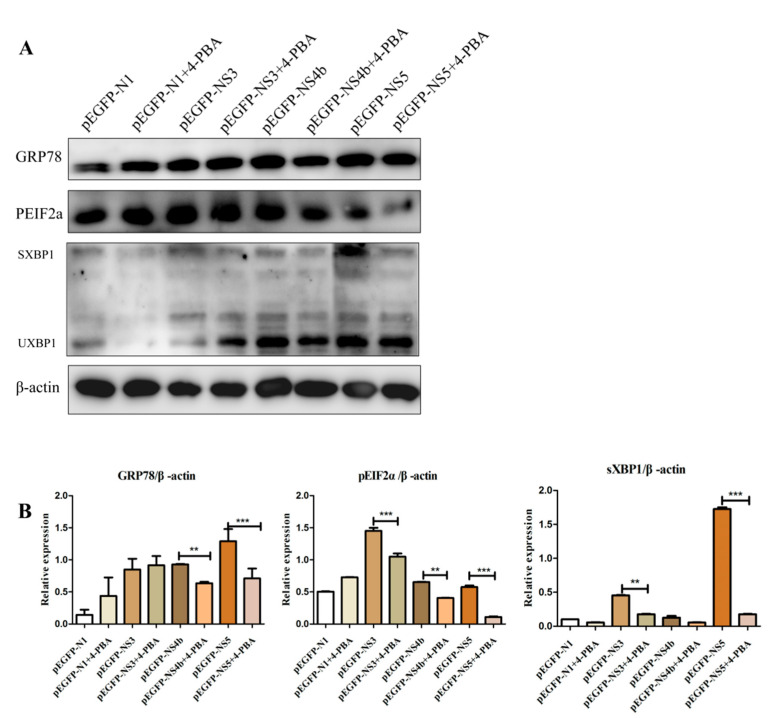
4-PBA inhibits the fusion protein of JEV to activate the IRE1 and PERK pathways. (**A**) The protein levels of GRP78, pEIF2α and sXBP1 in BV2 cells incubated with JEV (MOI 1.5) and 4-PBA of 1 mM/mL. (**B**) Densitometry scans of the GRP78, pEIF2α and sXBP1 brand of western blot from (**A**). ** *p* < 0.01; *** *p* < 0.001.

**Figure 5 viruses-15-00534-f005:**
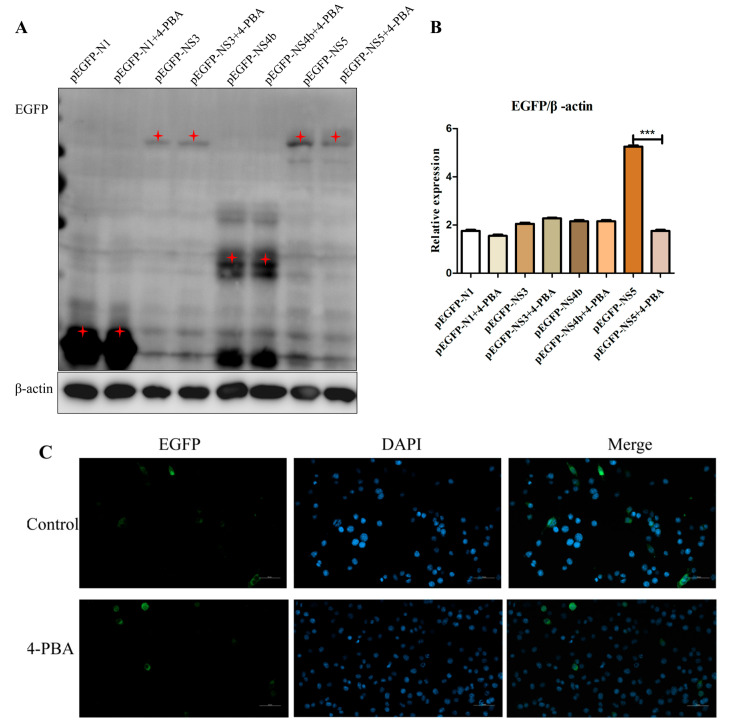
4-PBA inhibits NS5 expression in vitro. (**A**,**B**) Effect of 4-PBA on the expression of NS3, NS4b and NS5 proteins. (**C**) Laser confocal microscope detected NS5 gene expression. The red star is the correct size of protein. *** *p* < 0.001.

**Table 1 viruses-15-00534-t001:** primer used in the study.

Gene	Primer Sequence	Production
NS3	F:CGGCTAGCGCCACCATGGGGGGCGTGTTTTGGGACAC	1857 bp
	R: CCCTCGAGTCTCTTCCCTGCTGCAAAGTCTTTG	
NS4b	F:CGGCTAGCGCCACCATGAACGAGTACGGGATGCTAGA	765 bp
	R: CCCTCGAGCCTTTTCAAGGAGGGCTTGTC	
NS5 rdrp	F: CGGCTAGCGCCACCATGGAGGAAGATGTCAACCTAG	1920 bp
	R: CCCTCGAGGTATCTCCTGAGTGAAGTC	

The gene registration number is NC-001437.1.

**Table 2 viruses-15-00534-t002:** Primer sequences of real-time PCR.

Gene	Sequence	Product Length/bp	Accession
GRP78	F:TCGGACGCACTTGGAATGA	181 bp	NM_001163434.1
	R:GCCTCAGCAGTCTCCTTCA		
β-actin	F:GTGCTTCTAGGCGGACTGT	239 bp	NM_007393.5
	R:GCTCCAACCAACTGCTGTC		

## Data Availability

The datasets used and/or analyzed during the current study are available from the corresponding author upon reasonable request.

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
