# Peer review of "4-Phenyl-butyric Acid Inhibits Japanese Encephalitis Virus Replication via Inhibiting Endoplasmic Reticulum Stress Response"

_viruses, 2023, doi:10.3390/v15020534_

Round 1

Reviewer 1 Report

the overall concept of the study was good. The authors presented the experiments in a precise and concise manner. The appropriate statistical method was employed. The outcome will add to the scientific literature. 

Recommended for publication. 

Reviewer 2 Report

Reviewer’s concern

The study by Shuangshuang Wang et.al. is illustrating the impact of 4-PBA on the replication of JEV and hence it could be a useful chemotherapeutic agent for treating JEV. The study is compact , focused on one experimental variation and give conclusive results.

However, I have few concerns as below;

Authors need to work hard on improving their English language. Its hindering the communication of messages at many places. Sentences are losing its meaning and could be misinterpreted because of that.

Major points: 

1. In figure-1B, NS3 is higher in 4-PBA+ JEV lane. This seems opposite to the story that it inhibits JEV replication. Explain ?

2. The explanation of mechanism of this chemical would have added more value to the results. Authors might perform few more experiments to demonstrate so.

Minor points:

 1. In legend of figure 5, there are spelling mistakes.

  2. Overall language is not effective. Take help of some English professional.

Reviewer 3 Report

My major comments have been enclosed.
